# Triangle-Shaped Cerium Tungstate Nanoparticles Used to Modify Carbon Paste Electrode for Sensitive Hydroquinone Detection in Water Samples

**DOI:** 10.3390/s24020705

**Published:** 2024-01-22

**Authors:** Vesna Stanković, Slađana Đurđić, Miloš Ognjanović, Gloria Zlatić, Dalibor Stanković

**Affiliations:** 1Institute for Chemistry, Technology and Metallurgy, University of Belgrade, 11000 Belgrade, Serbia; vesna.stankovic@ihtm.bg.ac.rs; 2Faculty of Chemistry, University of Belgrade, 11000 Belgrade, Serbia; sladjanadj@chem.bg.ac.rs; 3Institute for Nuclear Science Vinča, University of Belgrade, 11000 Belgrade, Serbia; miloso@vin.bg.ac.rs; 4Faculty of Science and Education, University of Mostar, 88000 Mostar, Bosnia and Herzegovina; gloria.zlatic@fpmoz.sum.ba

**Keywords:** nanoparticle, modified electrode, hydroquinone, DPV, cerium, tungstate

## Abstract

In this study, we propose an eco-friendly method for synthesizing cerium tungstate nanoparticles using hydrothermal techniques. We used scanning, transmission electron microscopy, and X-ray diffraction to analyze the morphology of the synthesized nanoparticles. The results showed that the synthesized nanoparticles were uniform and highly crystalline, with a particle size of about 50 nm. The electrocatalytic properties of the nanoparticles were then investigated using cyclic voltammetry and electrochemical impedance spectroscopy. We further used the synthesized nanoparticles to develop an electrochemical sensor based on a carbon paste electrode that can detect hydroquinone. By optimizing the differential pulse voltammetric method, a wide linearity range of 0.4 to 45 µM and a low detection limit of 0.06 µM were obtained. The developed sensor also expressed excellent repeatability (RSD up to 3.8%) and reproducibility (RSD below 5%). Interferences had an insignificant impact on the determination of analytes, making it possible to use this method for monitoring hydroquinone concentrations in tap water. This study introduces a new approach to the chemistry of materials and the environment and demonstrates that a careful selection of components can lead to new horizons in analytical chemistry.

## 1. Introduction

Hydroquinone (HQ) is a colorless crystalline compound frequently used in the cosmetic industry as an additive in skin-lightening creams. However, its use in cosmetic products has been banned in many countries since 2001 due to its harmful effects on the skin. HQ has been proven to cause occupational vitiligo and exogenous ochronosis, both of which can cause discoloration and damage to the skin. Long-term exposure to HQ has also been linked to the development of carcinogenesis [1]. Although HQ is widely used in other industries, including medicine, pesticides, and photography, it is listed as a highly toxic pollutant by the United States Environmental Protection Agency (US EPA) due to its harmful impact on the environment and human health [2]. Different analytical methods are used to detect HQ in various environmental samples, including high-pressure liquid chromatography [3], mass spectrometry [4], and electrochemical methods [5]. The latest ones are based on the principle that HQ can be oxidized or reduced at an electrode surface, producing a measurable electrical current proportional to its concentration in the sample.

Electrochemical methods are popular for measuring a variety of analytes such as organic and inorganic pollutants [6,7,8,9], biologically active compounds [10,11], food additives [12], and pharmaceutical products [13,14,15] due to their simplicity, affordability, speed, and the possibility of miniaturization. However, to achieve satisfactory performance in terms of selectivity and sensitivity, the working electrodes used in these methods must be made of stable materials. The most commonly used electrodes are made of inert materials like silver, gold, and platinum, or carbon-based materials like glassy carbon, boron-doped diamond, and graphite [16]. Carbon paste electrodes (CPE) are particularly interesting due to their chemical inertness, robustness, reproducibility, stable response, and low ohmic resistance. Furthermore, CPE electrodes are non-toxic and environmentally friendly [17]. Although they have some disadvantages, such as lower sensitivity and reproducibility, slower kinetics of electron transfer, and lower stability in solutions, these issues can be addressed by modifying the carbon paste itself [18,19]. The surface of the working electrode can also be easily renewed, solving the issue of passivation [19], which makes these electrodes the most promising for developing new electrochemical sensors.

In recent years, increasing attention has been paid to nanoparticles of rare earth element (REE) compounds [20]. These elements have unique properties such as control structures, excellent physical and chemical properties, and a unique unpaired electronic configuration of 4f orbitals, making them exciting candidates for developing new electrochemical sensors [21,22]. REEs act as catalysts but also contribute to high conductivity and favorable optical and thermal properties [23,24]. Functional nanomaterials based on cerium, Ce (the most abundant REE), are characterized by high catalytic effects, thermal and chemical stability, semiconductor capabilities, large-scale oxygen storage potential, and eco-friendly properties [8,25]. Due to these outstanding performances, nano-size cerium-based compounds have found wide application as an integral part of the electrode material of various electrochemical sensors. Generally, researchers decide to synthesize functional electrode materials because their essential characteristics, such as size, shape, and morphology, directly affect the chemical interaction and selectivity toward the analyte. These properties can be further adjusted by changing the experimental conditions, depending on the purpose of the experiment. Also, the synthesis of materials allows multiple combinations of different components/compounds, which leads to the development of innovative (original) electrode modifiers. In addition, the starting components in material synthesis are usually significantly cheaper than commercially available electrode modifiers. Ma et al. have improved the catalytic performance of glassy carbon electrodes after modification with synthesized cerium-based nanocomposite (incorporated cerium vanadate into multi-walled carbon nanotubes) to determine the antibacterial drug sulfamethazine in water samples [26]. Manjula et al. synthesized cerium niobium oxide nanoparticles to the developed electrochemical sensor for p-nitrophenol in water samples [25], while Zhou et al. examined fungicide carbendazim levels in fruit, vegetable, and water samples using carbon cloth modified with synthesized cerium metal-organic framework [27]. Jandaghi et al. reported an electrochemical platform based on cerium-doped flower-shaped zinc oxide nanocrystallites to determine the cancer drugs methotrexate and epirubicin in pharmaceutical and clinical samples simultaneously [28]. At the same time, tungstate (WO_4_^2−^)-based materials have been the focus of world science in the last decade due to their exceptional magnetic, physiochemical, semiconductor, electrocatalytic, photocatalytic, and optical properties [29,30]. These materials have found wide applications in different industrial activities such as supercapacitors [31], lithium-ion batteries [32], and dye-sensitized solar cells [33]. The pronounced catalytic activity of the tungstate dominantly originates from the electrical conductivity of the tungsten atom (10^−4^ to 1 mS m^−1^) [29], which was additionally employed in photocatalytic degradation studies and electrochemical sensing. Pavithra et al. reported the sonochemical synthesis of nickel tungstate nanoparticles and their application in photocatalytic dye degradation and electrochemical determination of lead ions in environmental samples [34]. Mohammed et al. evaluated cadmium ion levels in water and urine samples using an electrochemical sensor based on nickel-tungstate-doped multi-walled carbon nanotube composites [35]. One-dimensional praseodymium tungstate nanoparticles at reduced graphene oxide nanocomposite for sensitive detection of insecticide fenitrothion in water samples is reported by Sundaresan et al. [36], while Goudarzi et al. described the development of a photocatalytic degradation platform based on eco-friendly synthesized manganese-doped thallium tungstate nanostructures for efficient removal of various antibiotics from environmental aquatic media [37].

This paper presents a method for producing cerium tungstate (Ce_2_(WO_4_)_3_) nanoparticles through hydrothermal synthesis. The study explores the use of these nanoparticles in modifying carbon paste electrodes (CPE) and presents the first investigation into the potential of cerium tungstate for this purpose. The resulting material was fully characterized both morphologically and electrochemically. The proposed sensor was then used to develop a method for detecting HQ, which was successfully applied to detect this analyte in drinking water samples. The developed sensor has a high potential for use in portable electrochemical workstations and can be adapted for detecting a variety of analytes in real time.

## 2. Materials and Methods

### 2.1. Materials and Apparatus

All chemicals used in this work were of analytical grade and obtained from Sigma Aldrich (St. Louis, MO, USA). Phosphoric acid (≥85 wt.% in H_2_O), acetic acid (glacial, ≥99%), and boric acid (≥99.5%), were used for the Britton Robinson buffer solutions (BRBS, 40 mM) preparation. A 1 M sodium hydroxide (NaOH) solution was used to adjust the pH value of the buffers.

A pH meter (Orion 1230, ThermoFisher Scientific, Waltham, MA, USA), equipped with a combined glass electrode (model Orion 9165BNWP, ThermoFisher Scientific, Waltham, MA, USA), was used for all pH measurements.

All electrochemical measurements—cyclic voltammetry, electrochemical impedance spectroscopy, and differential pulse voltammetry—were carried out on a potentiostat/galvanostat CHI760b (Bee Cave, TX, USA). The three-electrode system in the electrochemical cell was constructed using the Ag/AgCl (1 M KCl) reference electrode and platinum wire counter electrode (CH Instruments, Bee Cave, TX, USA), while the working electrode was a bare/modified carbon paste electrode (CPE).

Sample morphology was analyzed utilizing a JEM-2100F (Jeol, Tokyo, Japan) transmission electron microscope (TEM), operating at 200 kV, and a JEOL JSM-7001F (Jeol, Tokyo, Japan) scanning electron microscope (FE-SEM). The electron gun’s accelerating voltage was set to 20 kV to facilitate accurate quantitative EDS analysis.

The crystal structure of the prepared material underwent analysis via X-ray powder diffraction (XRPD). A high-resolution Smart Lab^®^ diffractometer (Rigaku, Tokyo, Japan) equipped with a Cu Kα radiation source (λ = 1.5406 Å) operating at 40 kV and 30 mA was used for measurements. Measurements were conducted on dried powder. Data collection involved a scanning pattern within the 10–70° 2θ range, with a scan rate of 0.5° per minute and a step size of 0.02° during the scan.

### 2.2. Preparation of Working Electrodes

For the preparation of Ce_2_(WO_4_)_3_ nanoparticles, we utilize a simple hydrothermal approach. Firstly, cerium nitrate (0.2 M, 15 mL) was dissolved in double-distilled water and stirred on a magnetic stirrer for 10 min. In the second beaker, sodium tungstate (0.2 M) was dissolved in 15 mL of double-distilled water, and in this solution, Triton-X 100 was added (0.4 g). After the stirring period, the second solution was added to a cerium nitrate solution under continuous stirring. The mixture was allowed to stir at room temperature for another 3 h. After this period, the obtained product was transferred to an autoclave (100 mL) and heated for 12 h at 160 °C. After a cooling period, the obtained product was centrifuged, washed several times with water and ethanol, and dried at 100 °C overnight. After that, the product was calcinated at 600 °C for three hours.

To prepare modified carbon paste electrodes, different amounts of Ce_2_(WO_4_)_3_ (mass percentages of 5%, 10%, and 15%) were used to evaluate the influence of synthesized nanoparticles on the electrochemical properties of the bare CPE. The carbon material was mixed with an appropriate amount of Ce_2_(WO_4_)_3_, and paraffin oil was added to the resulting mixture. The mixture was homogenized in a pestle and mortar for 30 min and then left for 24 h. After 24 h, the electrode was ready for the upcoming electrochemical measurements.

### 2.3. Electrochemical Measurements

Electrochemical investigations were carried out using cyclic voltammetry (CV), electrochemical impedance spectroscopy (EIS), and differential pulse voltammetry (DPV). The CV was performed to characterize the electrode and study HQ electrochemistry in the potential range from −0.4 to +1.2 V. DPV was used for electrochemical quantitative analysis of HQ in BRBS (pH 5). The potential range for DPV was from −0.1 V to +0.6 V with the following parameters: pulse amplitude of 0.01 V, pulse width of 0.2 s, sample width of 0.03 s, and pulse period of 0.5 s. To perform quantitative analysis, a calibration curve was constructed. During calibration, the appropriate amount of HK standard solution was added to the supporting electrolyte. The DPV was then recorded for each addition, and the value of the oxidation peak current was noted by tangent fit without background correction.

## 3. Results

### 3.1. Morphological Characterization of Ce_2_(WO_4_)_3_

The crystal structure of prepared nanoparticles was determined by analyzing X-ray powder diffraction (XRPD) data (Figure 1A). The diffraction patterns of the prepared sample were indexed in the monoclinic structure type (JCPDS No:96-2002722, space group C12/*c*1(15)) of cerium tungstate, Ce_2_(WO_4_)_3_, which matches well with the literature data for [18]. The XRPD data were also used to determine the crystallite size of cerium tungstate nanoparticles using the Scherrer equation (*D* = *Kʎ*/*cos* θ) applied to the most intense diffraction peaks. The average crystallite size was found to be (56 ± 5) nm.

Transmission electron microscope (TEM) images of synthesized Ce_2_(WO_4_)_3_ are shown in Figure 1B. It can be observed that the nanoparticles are shaped of a triangle similar to a guitar pick, with a particle size of about 50 nm. This, in accordance with the crystallite size obtained by XRPD, means that the particles are formed out of a single crystallite. Figure 1C shows a high-resolution image of a single nanoparticle, in which the ordered crystalline structure of the nanoparticle with clearly stacked planes can be observed.

The elemental mapping analysis of Ce_2_(WO_4_)_3_, taken from a wider sample area (350 µm^2^), reveals that nanoparticles consist of O, Ce, and W atoms with a relative share of 69.88% (O), 12.03% (Ce), and 18.09% (W) (Figure 1D). EDS analysis further affirms the uniformity of element distribution across the sample.

### 3.2. Electrochemical Characterization of Working Electrode

The bare and modified working electrodes were subjected to electrochemical examination using cyclic voltammetry (CV) and electrochemical impedance spectroscopy (EIS). Cyclic voltammetry is a widely used method for investigating the properties of redox systems, while EIS is a powerful technique for examining interface properties related to redox reactions that occur on the electrode surface itself. To study the effect of the modifier on the sensor’s properties, the carbon paste electrodes were modified with various mass fractions of Ce_2_(WO_4_)_3_ (5%, 10%, and 15%). Cyclic voltammetry was conducted in a 0.1 M KCl solution containing 5 mM [Fe(CN)_6_]^3+^/[Fe(CN)_6_]^4+^, using all prepared electrodes, and the corresponding voltammograms are presented in Figure 2A. As can be seen, the best electrochemical response was achieved using an electrode modified with 15% of Ce_2_(WO_4_)_3_, with the highest peaks currents, *I*p_a_ 74.7 µA and *I*p_c_—72.0 µA, while peak separation was found to be Δ*E*p 0.46 V. For electrodes modified with 5 and 10% of Ce_2_(WO_4_)_3_, lower peak currents were obtained (*I*p_a_ 62.7 µA, *I*p_c_—56.4 µA for 5% Ce_2_(WO_4_)_3_/CPE and *I*p_a_ 57.4 µA, *I*p_c_—64.6 µA for 10% Ce_2_(WO_4_)_3_/CPE), while peak-to-peak separations were 0.30 and 0.43 V for 5% Ce_2_(WO_4_)_3_/CPE and 10% Ce_2_(WO_4_)_3_/CPE, respectively. In comparison to the bare electrode (*I*p_a_ 48.5 µA, *I*p_c_—50.3 µA, and Δ*E*p 0.48 V), all modified electrodes have better electrochemical responses. These results indicate that the addition of the modifier delivers favorable effects on the characteristics of the CPE, independent of its mass fraction in the paste. This is likely attributed to the upgraded number of active sites and conductivity of the substance, which escalates correspondingly with the increase in Ce_2_(WO_4_)_3_, reaching its peak at a concentration of 15%.

Electrochemical impedance spectroscopy (EIS) is a robust technique that is used to explore the internal properties of electrode materials or specific processes that may impact an electrochemical system’s conductivity, resistance, and capacity. EIS is employed to investigate mass transfer, charge transfer, and diffusion processes in the electrode-solution system. The EIS spectra of bare and modified carbon paste electrodes were measured in a 0.1 M KCl solution containing 5 mM [Fe(CN)_6_]^3+^/[Fe(CN)_6_]^4+^. The frequency range was from 0.01 to 100,000 Hz. The impedance diagrams are shown in Figure 2B in the form of a Nyquist plot. All diagrams are characterized by two parts: the first part is a semi-circle in the high-frequency region, associated with a charge transfer at the electrode surface/solution interface, and characterized by a charge transfer resistance. The second is the straight line recorded at low frequency, which indicates a diffusion-controlled process at the electrode surface [19]. The electron transfer resistance (Rct) is obtained by EIS fitting to an equivalent circuit (inset of Figure 2B) and is estimated to be 12,900 Ω  for the bare CPE. This value dropped to 12,030 Ω for 5% Ce_2_(WO_4_)_3_/CPE and to 9465 Ω  for 10% Ce_2_(WO_4_)_3_/CPE. The lowest Rct value (7712 Ω) was obtained for CPE containing 15% of the modifier (Ce_2_(WO_4_)_3_). These results show that the value of the charge transfer resistance Rct is lowered after increasing modifier content, which demonstrates that the presence of Ce_2_(WO_4_)_3_ facilitates charge transfer at the interface. Based on these results, CPE modified with 15% of Ce_2_(WO_4_)_3_ was selected for further examination.

Electrokinetic properties of 15% Ce_2_(WO_4_)_3_/CPE were evaluated using cyclic voltammetry in a 0.1 M KCl solution containing 5 mM [Fe(CN)_6_]^3+^/[Fe(CN)_6_]^4+^, with varying the scan rate from 10 to 300 mVs^−1^. The corresponding CV curves are presented in Figure 2C. As can be seen, with an increase in scan rate, oxidation, and reduction, peak currents also increased. The linear dependence of peak currents versus the square root of scan rate (Figure 2D) indicates that the electrochemical process at the proposed electrode is diffusion-controlled. Randles–Sevcik equation: *I*p = 2:69 × 10^5^n^3/2^AD^½^ Cv^½^ (Eq. 1); where D is the diffusion coefficient of [Fe (CN)_6_]^3−/4−^ (cm^2^ s^−1^), Ip is the anodic/cathodic peak current (A), C is the concentration of the [Fe (CN)_6_]^3−/4−^ (mol cm^−3^), A is the electroactive area (cm^2^), n is the number of transferred electrons, and v^1/2^ is the square root of scan rate (mVs^−1^)^1/2^, was used to estimate the electroactive surface area of electrode. Based on the obtained results and calibration curve slope value, the electroactive surface area of the 15% Ce_2_(WO_4_)_3_/CPE modified electrode was found to be 0.063 cm^−2^, while the geometric surface area of the electrode is equal to 0.0314 cm^−2^. The incorporation of Ce_2_(WO_4_)_3_ nanoparticles in carbon paste has enhanced the properties of the modified electrode. Specifically, the modified electrode boasts a larger electroactive surface area and better conductive properties compared to a bare carbon paste electrode. This is due to the unique properties of utilized nanoparticles, which have been shown to enhance the conductivity and electroactivity of carbon paste electrodes.

Based on preliminary research, the CPE electrode modified with 15 mass percent of cerium tungstate has been shown to have the best electrochemical characteristics in terms of charge transfer. Therefore, this electrode was chosen for all further research. The next step was to examine the effect of the electrolyte pH value on the sensor response toward the target analyte. For this purpose, cyclic voltammograms of HQ with a concentration of 10 µM in solutions with different pH values (from 2 to 9) were recorded (Figure 3A). As the pH value of the supporting electrolyte increases, the analyte peak shifts towards more negative potential values, which indicates the existence of a strong dependence on the HQ redox reaction of the pH value—the redox reactions involve the protons. The dependence of the potential change and current intensities on the pH value are shown in Figure 3B,C. The anodic and cathodic peak potential is linearly proportional to pH values. The regression equations and corresponding slope values for *E*p_a_ vs. pH (0.061) and *E*p_c_ vs. pH (0.041), which are near the Nernst theoretical slope of 59 mV/pH, suggest that an equal number of protons and electrons are involved in the redox reaction of HQ. This observation is in accordance with other literature data and known electrochemical oxidation reactions for this kind of structure, where two protons and two electrons are included in the final formation of the quinone structure through the semiquinone step [20]. The possible electron transfer (ET) mechanism for the determination of HQ using Ce_2_(WO_4_)_3_ nanocomposite-immobilized CPE is presented in Figure 3D. For further development of the electrochemical method for HQ detection, a pH value of 5, was chosen based on peak shape and peak current intensities (Figure 3C).

The next step in the study was to analyze how the electrode response to HQ is affected by the scan rate. Cyclic voltammograms were recorded at various scanning speeds ranging from 10 to 280 mVs^−1^ in BRBS buffer solution (pH 5), containing 10 µM of HQ (Figure 3F). It was observed that as the scanning speed increased, the current of the oxidation peak HQ also increased. This increase in current is directly proportional to the square root of the scan rate (Figure 3F), indicating that the electrode process of hydroquinone oxidation at the electrode is diffusion-controlled.

### 3.3. Electrochemical Method for HQ Detection

Differential pulse voltammetry (DPV) has a considerably higher sensitivity and enhanced resolution than cyclic voltammetry. The DPV method was chosen to develop a detection method for HQ. By successively adding the appropriate amount of the standard HQ solution and recording the voltammogram after each addition, a calibration curve of the dependence of the current intensity on the analyte concentration was constructed (Figure 4A). All voltammograms were recorded in BRBS (pH 5) in triplicate, while the linear range was from 0.4 to 45 µM. The oxidation peak currents rise linearly with the concentration of the target molecule, and this linearity can be described using the following equation: I (A) = 3.92 × 10^−9^ C (μM)—2.78 × 10^−9^ with the correlation coefficient of R = 0.9983 (Figure 4B). The detection limit was calculated as 3 S/m (where S is the standard deviation of the blank and m is the calibration plot’s slope), which is 0.06 µM. All measurements were performed in triplicate.

It is essential to determine the selectivity of every analytical method developed for detecting a desired analyte. The proposed sensor against H*Q* detection was assessed for the influence of interfering substances by recording DPV voltammograms of analytes with and without interference in the BRBS solution (pH 5). The concentration ratio of the analyte: interference was 1:100. The study included interfering substances such as inorganic ions Na^+^, Mg^2+^, Ca^2+^, K^+^, Cl^−^, and AcO^−^ as well as some organic compounds like glucose (Glu), glycine (Gly), valine (Val), starch, ascorbic acid (AA), and oxalic acid (OA). Inorganic ions did not interfere with HQ determination. Among the organic compounds, there were no significant changes in HQ signal currents for all examined molecules, except for oxalic and ascorbic acids. In the presence of oxalic acid, the peak signal of HQ shifts slightly to a more positive value, while peak current intensities remain the same, probably due to a change in the pH value of the working solution. In the presence of AA, a notable peak current increase was observed.

The proposed sensor’s reproducibility was tested by creating five different carbon pastes, modified with 15% Ce_2_(WO_4_)_3_. These pastes were used to make electrodes, and DPV voltammograms of the BRBS (pH 5) containing 10 µM of HQ were recorded. The differences in anodic peak currents between the recordings did not exceed 5%, which shows that the sensor is highly consistent and reproducible.

To assess the method’s repeatability, the 15% Ce_2_(WO_4_)_3_/CPE was used as a working electrode to measure the same solution (10 µM HQ in BRBS buffer at pH 5) by DPV for ten successive measurements. The relative standard deviations of the peak currents were less than 3.8%, indicating excellent repeatability.

In Table 1, we presented some of the results of various research projects that have been conducted regarding the electrochemical detection of HQ. Based on the data presented in the table, it can be concluded that the outcomes obtained in this particular study are comparable to those that have been published so far. Although some sensors have been developed that offer smaller detection limits, it is worth noting that the preparation of the proposed sensor is relatively simple compared to the other sensors. Furthermore, it is important to highlight that the proposed sensor offers a practical solution for the detection of HQ, especially in the field of environmental monitoring, where the simplicity of preparation can be an advantage.

### 3.4. Real Sample Analysis

In order to evaluate the practical application of the sensor in the real world, a sample of drinking water was selected for testing. Initially, the sample was screened for HQ, but no HQ was detected. To further investigate, 10 µM of HQ was added to the sample three times in successive stages, and appropriate voltammograms were recorded after each addition. The results obtained (Table 2) and the achieved reliability values were found to be satisfactory, ranging from 97% to 106%. These results suggest that this sensor has the potential to be a useful tool for detecting HQ in real-world scenarios, including water samples. The findings of this study demonstrate the sensor’s ability to perform accurate and reliable measurements of HQ levels, indicating its potential for widespread use in various applications where HQ detection is necessary.

## 4. Conclusions

This paper presents a straightforward method to synthesize cerium tungstate nanoparticles through hydrothermal synthesis. The nanoparticles obtained were utilized for modifying carbon paste electrodes, which were then characterized electrochemically and morphologically. The modified sensor demonstrated improved performance compared to the bare CPE, owing to the modifier’s contribution to better electron transfer. The sensor was tested for the detection of hydroquinone as an analyte of interest, exhibiting a wide linear range (0.4–45 µM), a low detection limit (0.06 µM), and satisfactory selectivity and reproducibility. It is suitable for detection applications and can be conveniently used in portable electrochemical workstations. The principle of this sensor can be extended to a broader range of analytes, making it reliable and efficient for real-time detection.

## Figures and Tables

**Figure 1 sensors-24-00705-f001:**
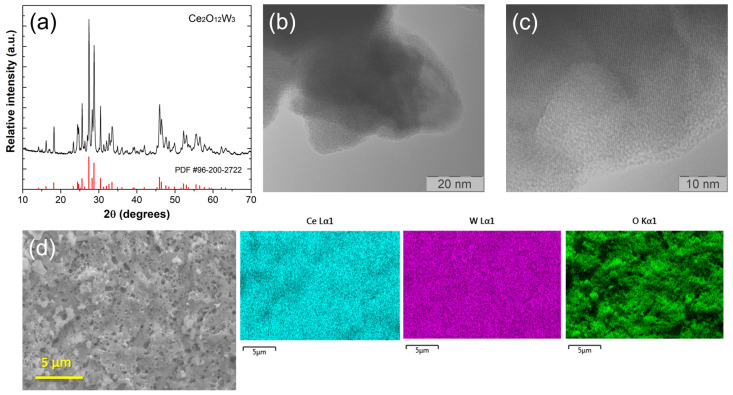
(**a**) XRPD diffractogram of Ce_2_(WO_4_)_3_ nanoparticles, (**b**,**c**) HR-TEM micrographs of prepared nanoparticles, and (**d**) FE-SEM micrograph of wider sample area and corresponding EDS elemental mapping of Ce, W, and O, respectively.

**Figure 2 sensors-24-00705-f002:**
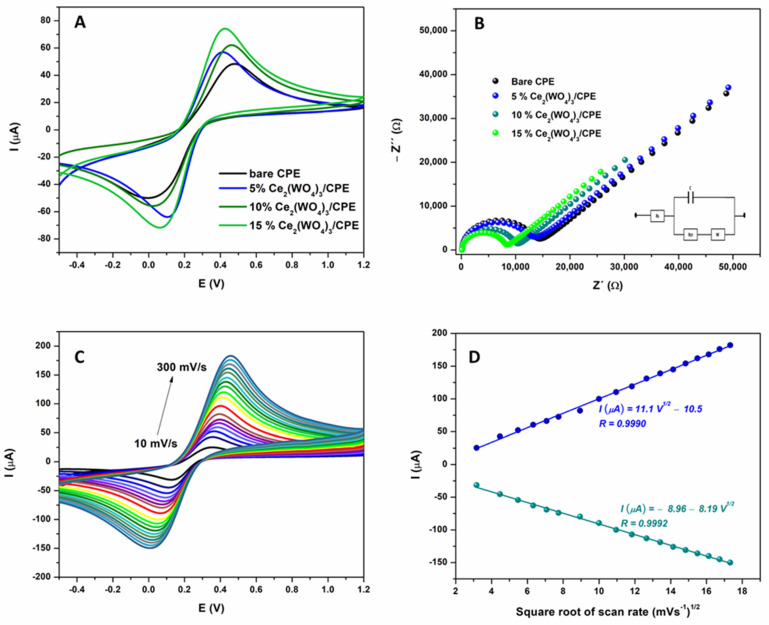
(**A**) CV and (**B**) EIS diagrams of 5 mM [Fe(CN)_6_]^3−^/[Fe(CN)_6_]^4−^ in 0.1 KCl, obtained by using bare CPE and CPE modified with 5, 10, and 15% of Ce_2_(WO_4_)_3_. (**C**) CV in 0.1 M KCl containing 5 mM [Fe(CN)_6_] ^3−^/[Fe(CN)_6_]^4−^, scan rate range 10 to 300 mVs^−1^ with 15% Ce_2_(WO_4_)_3_/CPE as working electrode, with (**D**) corresponding linear relationship of peak currents vs. square root of scan rate.

**Figure 3 sensors-24-00705-f003:**
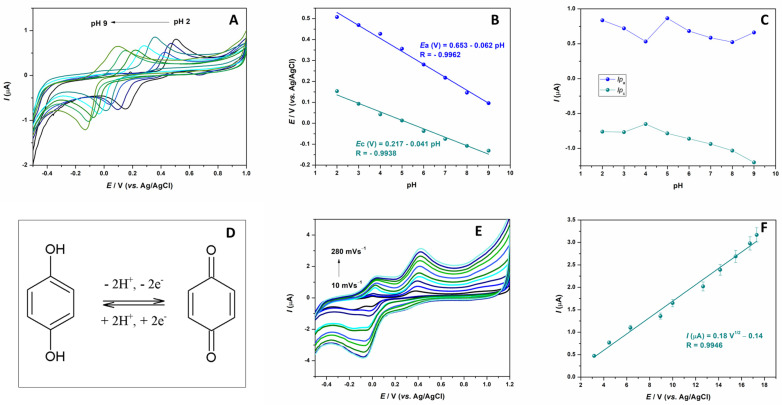
(**A**) CVs of 10 µM HQ in BRBS in pH range from 2 to 9; (**B**) Dependence of peaks potential (*E*p_a_ and *E*p_c_) and (**C**) peaks currents (*I*p_a_ and *I*p_c_) of pH value of supporting solution (**D**) CVs of 10 µM HQ in BRBS, pH 5, in scan rate range from 10 to 280 mVs^−1^ (**E**) Linear relationship of signal peak currents (*I*p_a_) vs. square root of scan rate (**F**) Proposed redox reaction of HQ on 15% Ce_2_(WO_4_)_3_/CPE.

**Figure 4 sensors-24-00705-f004:**
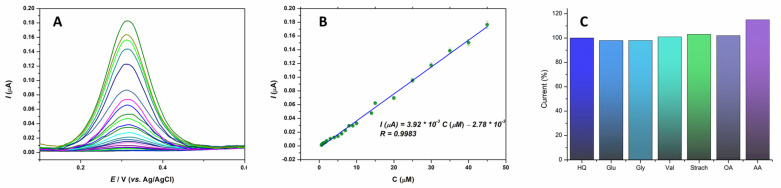
(**A**) Calibration curves for successive addition of HQ in BRBS (pH 5) using 15% Ce_2_(WO_4_)_3_/CPE, (**B**) Linear relationship between peak currents and HQ concentration, and (**C**) Influence of interfering compound on HQ signal.

**Table 1 sensors-24-00705-t001:** Performance of reported electrochemical sensors for the determination of HQ.

	Linear Range (µM)	LOD (µM)	Technique	Reference
Ce_2_(WO_4_)_3_/CPE	0.4–45	0.06	DPV	Our work
MnOx/rGO/SPE	0.5–200	0.049	DPV	[5]
MgO/GO/MCPE	10–80	0.37	CV	[38]
NiO/CNT/GCE	10–500	2.5	DPV	[39]
Pd@TiO_2_–SiC nanohybrid GCE	0.01–200	0.0055	DPV	[40]
Mn_2_SnO_4_/*f*-CB/SPCE sensor	0.005–70.80	0.007	LSV	[41]
ZnO/Co_3_O_4_	10–100	3.226	DPV	[42]
Co_3_O_4_/CSs-GCE	2–60	0.0026	DPV	[43]
GCE@Co–SnO_2_–PANI	0.02–0.2	0.00494	DPV	[44]
CoNi-MOF/GO/GCE	0.1–100	0.03	DPV	[45]
CoWO_4_@GCE	0.02–0.32	0.00221	Amperometry	[46]

**Table 2 sensors-24-00705-t002:** Real sample analysis.

Sample	HQ Addition (µM)	HQ Found (µM)	Recovery (%)
Water	10.00	10.58	105.8
	10.00	21.24	106.2
	10.00	29.65	98.8

## Data Availability

Data are contained within the article.

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
