# Peer review of "Triangle-Shaped Cerium Tungstate Nanoparticles Used to Modify Carbon Paste Electrode for Sensitive Hydroquinone Detection in Water Samples"

_sensors, 2024, doi:10.3390/s24020705_

Round 1
Reviewer 1 Report
Comments and Suggestions for Authors
In this paper, the authors synthesis cerium tungstate nanoparticles for Electrochemical Behavior of hydroquinone Sensing Applications The synthesized nanoparticles are well characterized and perform well. Before publication is considered, the level of English needs to be improved. Detailed below are additional suggestions that should be considered when submitting the revised manuscript.
1. Please redesign the abstract section to be more readable by focusing on the merit of figures.
2. In introduction, the materials part should be discussed more. Please discuss issues and challenge in previous electrocatalyst, why author go for the synthesis nanoparticles, how it is unique from other catalyst, and novelty of this work should be included.
3. Essential related work may be cited if appropriate 1. Process Safety and Environmental Protection-171, (705-716); Colloids and Surfaces A: Physicochemical and Engineering Aspects- 653, (129941); Process Safety and Environmental Protection- 165, (151-160)
4. What happens to the activity of the electrode when you further increase the mass percentage to 20 and 25%?
5. Please use the same unit throughout the manuscript, (i.e in Fig. 2a author use to represent current in A, further in B used to represents in µA.
6. Authors are suggested to include a comparative table for the usage of other metal oxide materials for the detection of HQ and list out the different techniques used for the detection of HQ with its minimal detection limits.
7. The electrochemical mechanism of HQ on the prepared electrode should be discussed elaborately.
8. For the repeatability test how many times other performed the electrodes? The discussion should be elaborate.
9. In addition to above points, a thorough inspection is required, because many typographical, grammatical errors or poor English are distributed in the manuscript, the manuscript should be carefully checked, and necessary corrections should be done.
Comments on the Quality of English Language
poor English are distributed in the manuscript, the manuscript should be carefully checked, and necessary corrections should be done.
Author Response
Reviewer 1
In this paper, the authors synthesis cerium tungstate nanoparticles for Electrochemical Behavior of hydroquinone Sensing Applications The synthesized nanoparticles are well characterized and perform well. Before publication is considered, the level of English needs to be improved. Detailed below are additional suggestions that should be considered when submitting the revised manuscript.
We would like to express our sincere gratitude to the reviewer for taking the time to provide us with valuable feedback. We appreciate the kind compliment as well as the insightful suggestions provided in the review.
We have taken the reviewer's comments into consideration and carefully examined each question raised. As a result, we made the necessary revisions to the text, ensuring that all concerns were adequately addressed.
- Please redesign the abstract section to be more readable by focusing on the merit of figures.
Thank you very much for this suggestion. We rewrite the Abstract section as follow:
In this study, we propose an eco-friendly method for synthesizing cerium tungstate nanoparticles using hydrothermal techniques. We used scanning, transmission electron microscopy, and X-ray diffraction to analyze the morphology of the synthesized nanoparticles. The results showed that the synthesized nanoparticles were uniform and highly crystalline, with a particle size of about 50 nm. The electrocatalytic properties of the nanoparticles were then investigated using cyclic voltammetry and electrochemical impedance spectroscopy. We further used the synthesized nanoparticles to develop an electrochemical sensor based on a carbon paste electrode that can detect hydroquinone. By optimizing the differential pulse voltammetric method, a wide linearity range of 0.4 to 45 µM and a low detection limit of 0.06 µM were obtained. The developed sensor also expressed excellent repeatability (RSD up to 3.8%) and reproducibility (RSD below 5%). Interferences had an insignificant impact on the determination of analytes, making it possible to use this method for monitoring hydroquinone concentration in tap water. This study introduces a new approach to the chemistry of materials and the environment and demonstrates that a careful selection of components can lead to new horizons in analytical chemistry.
- In introduction, the materials part should be discussed more. Please discuss issues and challenge in previous electrocatalyst, why author go for the synthesis nanoparticles, how it is unique from other catalyst, and novelty of this work should be included.
Thank you very much for this kindly suggestion. We have incorporated the reviewer's feedback to improve the clarity and comprehensiveness of the section. Our revisions include a more detailed discussion of the reasons and needs for synthesizing nanoparticles, with an emphasis on the unique properties and potential applications of rare earth-based nanoparticles, particularly those containing cerium. We rewrite Introduction section as follow:
REEs act as a catalyst, but also contribute to high conductivity and favorable optical and thermal properties [23,24]. Functional nanomaterials based on cerium, Ce (a most abundant REE) are characterized by high catalytic effect, thermal and chemical stability, semi-conductor capabilities, large-scale oxygen storage potential and eco-friendly properties [8,25]. Due to these outstanding performances, nano-size cerium-based compounds have found wide application as an integral part of the electrode material of various electro-chemical sensors. Generally, researchers decide to synthesize functional electrode materials because their essential characteristics, such as size, shape and morphology, directly affect the chemical interaction and selectivity toward the analyte. These properties can be further adjusted by changing the experimental conditions depending on the purpose of the experiment. Also, the synthesis of materials allows multiple combinations of different components/compounds, which leads to the development of innovative (original) electrode modifiers. In addition, the starting components in material synthesis are usually significantly cheaper than commercially available electrode modifiers. Ma et al. have improved the catalytic performance of glassy carbon electrodes after modification with synthesized cerium-based nanocomposite (incorporated cerium vanadate into multi-walled carbon nanotubes) to determine the antibacterial drug sulfamethazine in water samples [26]. Manjula et al. synthesized cerium niobium oxide nanoparticles to the developed electrochemical sensor for p-nitrophenol in water samples [25], while Zhou et al. examined fungicide carbendazim levels in fruit, vegetable and water samples using carbon cloth modified with synthesized cerium metal-organic framework [27]. Jandaghi et al. re-ported an electrochemical platform based on cerium-doped flower-shaped zinc oxide nano-crystallites to determine the cancer drugs methotrexate and epirubicin in pharmaceutical and clinical samples simultaneously [28]. At the same time, tungstate (WO42-)-based materials have been the focus of world science in the last decade due to their exceptional magnetic, physiochemical, semiconductor, electrocatalytic, photocatalytic and optical properties [29,30]. These materials have found wide application in different industrial activities such as supercapacitors [31], lithium-ion batteries [32] and dye-sensitized solar cells [33]. The pronounced catalytic activity of tungstate dominantly originates from the electrical conductivity of tungsten atom (10-4 to 1 mS/m) [29], which was additionally employed in photocatalytic degradation studies and electrochemical sensing. Pavithra et al. reported the sonochemical synthesis of nickel tungstate nanoparticles and their application in photocatalytic dye degradation and electrochemical determination of lead ions in environmental samples [34]. Mohammed et al. evaluated cadmium ion levels in water and urine samples using an electrochemical sensor based on nickel tungstate-doped multi-walled carbon nanotube composites [35]. One‑dimensional praseodymium tungstate nanoparticles@reduced graphene oxide nanocomposite for sensitive detection of insecticide fenitrothion in water samples is reported by Sundaresan et al. [36], while Goudarzi et al. described the development of a photocatalytic degradation platform based on eco-friendly synthesized manganese-doped thallium tungstate nanostructures for efficient removal of various antibiotics from environmental aquatic media [37].
This paper presents a method for producing cerium tungstate (Ce2(WO4)3) nanoparticles through hydrothermal synthesis. The study explores the use of these nanoparticles in modifying carbon paste electrodes (CPE) and presents the first investigation into the potential of cerium tungstate for this purpose. The resulting material was fully characterized both morphologically and electrochemically. The proposed sensor was then used to develop a method for detecting HQ, which was successfully applied to detect this analyte in drinking water samples. The developed sensor has high potential for use in portable electrochemical workstations and can be adapted for detecting a variety of analytes in real-time.
- Essential related work may be cited if appropriate 1. Process Safety and Environmental Protection-171, (705-716); Colloids and Surfaces A: Physicochemical and Engineering Aspects- 653, (129941); Process Safety and Environmental Protection- 165, (151-160)
Thank you for this suggestion. We include proposed references in manuscript
- What happens to the activity of the electrode when you further increase the mass percentage to 20 and 25%?
Thank you for bringing up the important point about the optimum amount of catalyst in carbon paste. We understand that the original manuscript did not provide enough details on how the catalyst content affects the electrode response of the sensor. Namely, if the catalyst loading is less than 5 wt.%, there may not be enough active material to utilize the high surface area of CPE, which can limit the number of electrochemically active sites for redox reactions. On the other hand, as the catalyst loading increases, more active sites become available, leading to higher redox peak currents. However, if the catalyst loading surpasses 20 wt.%, catalyst particles may start to aggregate, which can reduce the exposed surface area of the catalyst and limit the electrolytes' access to active sites. This can lead to low reproducibility of measurements. The reason for this behavior is likely due to the difficulty of preparing a uniform paste, resulting from the aggregation of catalyst particles.
- Please use the same unit throughout the manuscript, (i.e in Fig. 2a author use to represent current in A, further in B used to represents in µA.
Thanks for carefully observing the work. We standardized the units of current intensity to microamperes (µA) and double-checked all figures.
- Authors are suggested to include a comparative table for the usage of other metal oxide materials for the detection of HQ and list out the different techniques used for the detection of HQ with its minimal detection limits.
Thank you for this suggestion. We include proposed table in manuscript, and short remark about comparation results as follow:
In Table 1, we presented some of the results of various research works that have been conducted regarding the electrochemical detection of HQ. Based on the data presented in the table, it can be concluded that the outcomes obtained in this particular study are comparable to those that have been published so far. Although some sensors have been developed that offer lower detection limits, it is worth noting that the preparation of the proposed sensor is relatively simple compared to the other sensors. Furthermore, it is important to highlight that the proposed sensor offers a practical solution for the detection of HQ, especially in the field of environmental monitoring, where the simplicity of preparation can be an advantage.
- The electrochemical mechanism of HQ on the prepared electrode should be discussed elaborately.
Thank you for this suggestion. We include the possible electron transfer (ET) mechanism for determination of HQ using Ce2(WO4)3 nanocomposite immobilized CPE in manuscript, with the existing explanation, as follow:
The anodic and cathodic peak potential is linearly proportional to pH values. The regression equations and corresponding slope values for Epa vs. pH (0.061) and Epc vs. pH (0.041), which are near to the Nernst theoretical slope of 59 mV/pH, suggest that equal number of protons and electrons are involved in the redox reaction of HQ. This observation is in accordance with other literature data and known electrochemical oxidation reaction for this kind of structure, where two protons and two electrons are included in the final formation of quinone structure through semiquinone step [20]. The possible electron transfer (ET) mechanism for determination of HQ using Ce2(WO4)3 nanocomposite immobilized CPE is presented in Figure 2F.
- For the repeatability test how many times other performed the electrodes? The discussion should be elaborate.
We added missing information in manuscript and rephrase this section as follow:
The proposed sensor's reproducibility was tested by creating five different carbon pastes, modified with 15% Ce2(WO4)3. These pastes were used to make electrodes, and DPV voltammograms of the BRBS (pH 5), containing 10 µM of HQ, were recorded. The differences in anodic peak currents between the recordings did not exceed 5%, which shows that the sensor is highly consistent and reproducible.
To assess the method's repeatability, the 15% Ce2(WO4)3/CPE was used as working electrode to measure the same solution (10 µM HQ in BRBS buffer at pH 5) by DPV for ten successive measurements. The relative standard deviations of the peak currents were less than 3.8%, indicating excellent repeatability.
- In addition to above points, a thorough inspection is required, because many typographical, grammatical errors or poor English are distributed in the manuscript, the manuscript should be carefully checked, and necessary corrections should be done.
Thank you for these suggestions. A native English speaker read the text and made appropriate corrections
Comments on the Quality of English Language
poor English are distributed in the manuscript, the manuscript should be carefully checked, and necessary corrections should be done.
Thank you for these suggestions. A native English speaker read the text and made appropriate corrections
Reviewer 2 Report
Comments and Suggestions for Authors
The manuscript presents a cerium tungstate modified carbon paste electrode for the determination o hydroquinone in water samples. The paper presents a good overall quality although some aspects need improvement before considering its publication.
Introduction:
In sentence: “Different analytical methods are used to detect HQ in various environmental samples, including high-pressure liquid chromatography, mass spectrometry, and electrochemical methods”, please include references for the cited analytical methods.
Experimental
Please describe how the calibration curve was performed.
Results and discussion
All figures containing as axis E and I should be in italic.
Please add the reference electrode to the potential axis when presented e.g. E / V (vs. Ag/AgCl)
In discussion, the authors state: “The impedance diagrams are shown in Figure 2B in the form of Nyquist plots. All diagrams are characterized by two parts: the first part is a semi-circle in the high frequency region, associated with a charge transfer at the electrode surface/solution inter- face and characterized by a charge transfer resistance. The second is the straight line recorded at low frequency which indicates a diffusion-controlled process at the electrode surface [19]. The electron transfer resistance (Rct) is estimated to be 14700 Ω for the bare CPE. This value dropped to 13500 Ω for 5% Ce2(WO4)3/CPE and to 10350 Ω for 10 % Ce2(WO4)3/CPE. The lowest Rct value (8540 Ω) was obtained for CPE containing 15 % of modifier (Ce2(WO4)3).”
I)The diagram presented is the Nyquist plot. Please name it accordingly.
II) How did the authors obtain the RCT values? By EIS fitting to an equivalent circuit or graphically? Please specify and detail how these values were obtained.
III) The straight line behavior in low frequencies does not ensure that is a diffusion-controlled process. The angle of the straight line must be 45 degrees to correspond to a diffusion-controlled process or the spectra needs to be adjusted to a Warburg impedance. Please add the phase angle to the discussion or present the equivalent circuit used to fit the spectra if so.
In page 5 the authors state: Based on obtained results and calibration curve slope value, the electroactive surface area of 15 % Ce2(WO4)3 /CPE modified electrode was found to be 0.063 cm-2 . How does that compare to the geometric area of the electrode? Please inform the geometric area and add a discussion involving it.
In HQ determination, there are errors bars in the graphics but the authors did not explicit the number of measurement repetitions. Was the curve obtained in triplicate?
Include that information in the reproducibility and repeatability experiments as well.
How does your sensor compare to other HQ sensors? Please add a brief discussion that puts the proposed sensor in context.
In table 1 find is an irregular verb therefore the past perfect tense iis found not founded.
Please revise the English of the manuscript.
In real sample analysis the authors state: In order to evaluate the practical application of the sensor in real-world, a sample of 266 drinking water was selected for testing.
Please provide a geographic location where the sample was obtained.
Comments on the Quality of English LanguageThe overall english quality is satisfactory needing some minor revision.
Author Response
Reviewer 2
The manuscript presents a cerium tungstate modified carbon paste electrode for the determination of hydroquinone in water samples. The paper presents a good overall quality although some aspects need improvement before considering its publication.
We would like to express our sincere gratitude to the reviewer for taking the time to provide us with valuable feedback. We appreciate the kind compliment as well as the insightful suggestions provided in the review.
We have taken the reviewer's comments into consideration and carefully examined each question raised. As a result, we made the necessary revisions to the text, ensuring that all concerns were adequately addressed.
Introduction:
- In sentence: “Different analytical methods are used to detect HQ in various environmental samples, including high-pressure liquid chromatography, mass spectrometry, and electrochemical methods”, please include references for the cited analytical methods.
Thank you for this suggestion. We include appropriate references in manuscript.
Experimental
- Please describe how the calibration curve was performed.
Thank you for this suggestion. We include appropriate paragrafs in Experimental section as follow:
Electrochemical measurements
Electrochemical investigations were carried out using cyclic voltammetry (CV), electrochemical impedance spectroscopy (EIS) and differential pulse voltammetry (DPV). The CV was performed to characterize the electrode and study HQ electrochemistry in the potential range from −0.4 to +1.2 V. DPV was used for electrochemical quantitative analysis of HQ in BRBS (pH 5). The potential range for DPV was from -0.1 V to +0.6 V with the following parameters: pulse amplitude of 0.01 V, pulse width of 0.2 s, sample width of 0.03 s and pulse period of 0.5 s. To perform quantitative analysis, a calibration curve was constructed. During calibration, the appropriate amount of HK standard solution was added to the supporting electrolyte. The DPV was then recorded for each addition, and the value of the oxidation peak current was noted by tangent fit without background correction.
Results and discussion
- All figures containing as axis Eand I should be in italic.
Thank you for this suggestion. All figures that contained E and I have been corrected according to the comment.
- Please add the reference electrode to the potential axis when presented e.g. E/ V (vs. Ag/AgCl)
Thank you for this suggestion. We made appropriate changes in manuscript and in all figures.
- In discussion, the authors state: “The impedance diagrams are shown in Figure 2B in the form of Nyquist plots. All diagrams are characterized by two parts: the first part is a semi-circle in the high frequency region, associated with a charge transfer at the electrode surface/solution inter- face and characterized by a charge transfer resistance. The second is the straight line recorded at low frequency which indicates a diffusion-controlled process at the electrode surface [19]. The electron transfer resistance (Rct) is estimated to be 14700 Ω for the bare CPE. This value dropped to 13500 Ω for 5% Ce2(WO4)3/CPE and to 10350 Ω for 10 % Ce2(WO4)3/CPE. The lowest Rct value (8540 Ω) was obtained for CPE containing 15 % of modifier (Ce2(WO4)3).”
I)The diagram presented is the Nyquist plot. Please name it accordingly.
Thank you for this remark. We corrected this.
- II) How did the authors obtain the RCTvalues? By EIS fitting to an equivalent circuit or graphically? Please specify and detail how these values were obtained.
Thank you for this kindly suggestion. We performed EIS fitting to an equivalent circuit and based on these results we obtained the RCT values. We added this information in manuscript as follow:
The electron transfer resistance (Rct) is obtained by EIS fitting to an equivalent circuit (in-set of Figure 2B) and is estimated to be 12900 Ω for the bare CPE. This value dropped to 12030 Ω for 5% Ce2(WO4)3/CPE and to 9465 Ω for 10 % Ce2(WO4)3/CPE. The lowest Rct value (7712 Ω) was obtained for CPE containing 15 % of modifier (Ce2(WO4)3).
III) The straight line behavior in low frequencies does not ensure that is a diffusion-controlled process. The angle of the straight line must be 45 degrees to correspond to a diffusion-controlled process or the spectra needs to be adjusted to a Warburg impedance. Please add the phase angle to the discussion or present the equivalent circuit used to fit the spectra if so.
Once again, thank you for this kindly suggestion. We performed EIS fitting to an equivalent circuit and based on these results we obtained the RCT values. We present the equivalent circuit used to fit the spectra.
- In page 5 the authors state: Based on obtained results and calibration curve slope value, the electroactive surface area of 15 % Ce2(WO4)3 /CPE modified electrode was found to be 0.063 cm-2. How does that compare to the geometric area of the electrode? Please inform the geometric area and add a discussion involving it.
Thank you for this suggestion. We include appropriate information and discussion in the main text as follow:
Based on obtained results and calibration curve slope value, the electroactive surface area of 15 % Ce2(WO4)3/CPE modified electrode was found to be 0.063 cm-2, while the geometric surface area of the electrode is equal to 0.0314 cm-2. The incorporation of Ce2(WO4)3 nanoparticles in carbon paste has enhanced the properties of the modified electrode. Specifically, the modified electrode boasts a larger electroactive surface area and better conductive properties compared to a bare carbon paste electrode. This is due to the unique properties of utilized nanoparticles, which have been shown to enhance the conductivity and electroactivity of carbon paste electrodes.
- In HQ determination, there are errors bars in the graphics but the authors did not explicit the number of measurement repetitions. Was the curve obtained in triplicate?
Yes, it is. We are grateful for this observation and we have added missing information in manuscript.
- Include that information in the reproducibility and repeatability experiments as well.
We added missing information in manuscript and rephrase this section as follow:
The proposed sensor's reproducibility was tested by creating five different carbon pastes, modified with 15% Ce2(WO4)3. These pastes were used to make electrodes, and DPV voltammograms of the BRBS (pH 5), containing 10 µM of HQ, were recorded. The differences in anodic peak currents between the recordings did not exceed 5%, which shows that the sensor is highly consistent and reproducible.
To assess the method's repeatability, the 15% Ce2(WO4)3/CPE was used as working electrode to measure the same solution (10 µM HQ in BRBS buffer at pH 5) by DPV for ten successive measurements. The relative standard deviations of the peak currents were less than 3.8%, indicating excellent repeatability.
- How does your sensor compare to other HQ sensors? Please add a brief discussion that puts the proposed sensor in context.
Thank you for this suggestion. We include comparation table in manuscript, and short remark about comparation results as follow:
In Table 1, we presented some of the results of various research works that have been conducted regarding the electrochemical detection of HQ. Based on the data presented in the table, it can be concluded that the outcomes obtained in this particular study are comparable to those that have been published so far. Although some sensors have been developed that offer smaller detection limits, it is worth noting that the preparation of the proposed sensor is relatively simple compared to the other sensors. Furthermore, it is important to highlight that the proposed sensor offers a practical solution for the detection of HQ, especially in the field of environmental monitoring, where the simplicity of preparation can be an advantage.
- In table 1 find is an irregular verb therefore the past perfect tense iis found not founded. Please revise the English of the manuscript.
Thank you for these suggestions. A native English speaker read the text and made appropriate corrections
- In real sample analysis the authors state: In order to evaluate the practical application of the sensor in real-world, a sample of 266 drinking water was selected for testing. Please provide a geographic location where the sample was obtained.
Thank you for this remark. This is probably a technical error. Namely, in this research, a sample of drinking water was used as a real sample, for testing the practical application of the method.
Comments on the Quality of English Language
The overall English quality is satisfactory needing some minor revision.
Thank you for these suggestions. A native English speaker read the text and made appropriate corrections
Reviewer 3 Report
Comments and Suggestions for Authors
Title : ok if you can make it shorter is better
Abstract : Ok but add some pourcentages about the detection of the hydroquinone and reproductibilities?
Introduction: HQ is banned since 2001, what is the necessery or the importance of your study ? can we use your approch for other chemicals compounds.
In my opinion this paper can be accepted if the is other compounds which can be used to detect insteade of HQ?
Author Response
Reviewer 3
- Title : ok if you can make it shorter is better
Thank you for this remark. We rewrite Title as follow:
Triangle-shaped cerium tungstate nanoparticles used to modify carbon paste electrode for sensitive hydroquinone detection in water samples
- Abstract : Ok but add some pourcentages about the detection of the hydroquinone and reproductibilities?
Thank you very much for this suggestion. We added some additional information and rewrite the Abstract section as follow:
In this work, we propose a simple hydrothermal synthesis of cerium tungstate nanoparticles, following the principles of green chemistry. The morphology of the obtained sample was examined using scanning and transmission electron microscopy and X-ray diffraction, which reveals uniform and highly crystalline structure of synthesized nanoparticles, with a particle size of about 50 nm. The electrocatalytic properties were investigated using cyclic voltammetry and electrochemical impedance spectroscopy. The material further served to develop an electrochemical sensor based on a carbon paste electrode to detect hydroquinone. By carefully optimizing the differential pulse voltammetric method, a wide range of linearity from 0.4 to 45 µm and low detection limit of 0.06 µm was obtained. The developed sensor expressed excellent repeatability (RSD up to 3.8%) and reproducibility (RSD bellow 5%) The insignificant influence of interferences on the determination of analytes resulted in the successful application of the method for monitoring the concentration of hydroquinone in tap water. This method provides new approaches in the chemistry of materials and the environment, but it is basically electroanalytical, which shows that a careful selection of components can open new horizons in the field of analytical chemistry.
- Introduction: HQ is banned since 2001, what is the necessery or the importance of your study ? can we use your approch for other chemicals compounds.
Thank you for your question. Hydroquinone is prohibited for use in cosmetics, but it is still being sold illegally or under different brand names without being mentioned on the label. This makes it important to develop an affordable and easily accessible method for detecting hydroquinone, particularly in developing countries where regulation and control of cosmetic products are not yet adequate.
- In my opinion this paper can be accepted if the is other compounds which can be used to detect insteade of HQ?
Thank you very much for this comment. This study focused on synthesizing cerium tungstate nanoparticles and evaluating their suitability for modifying carbon paste. The cerium tungstate nanoparticles were used for the first time to modify carbon paste, which then acts as a working electrode for the detection of hydroquinone (HQ), a common pollutant in water. This study demonstrates the potential of cerium tungstate nanoparticles for use in modifying carbon paste and developing electrochemical sensors for different chemical compounds. The findings of this study can contribute to the development of more efficient and reliable sensors for environmental monitoring and biomedical applications.
Round 2
Reviewer 1 Report
Comments and Suggestions for Authors
The Manuscript has been revised according to the reviewer's suggestion. Therefore, I request editor to accept the manuscript for publication in your esteemed journal.
Comments on the Quality of English LanguageThe quality of English language has improved.
Reviewer 3 Report
Comments and Suggestions for Authors
The authors have performed all the recommandations
in my opinion this work can be accepted in the revised form
with regards